# Primary care-based screening and management of depression amongst heavy drinking patients: Interim secondary outcomes of a three-country quasi-experimental study in Latin America

Amy O'Donnell[1]*, Bernd Schulte[2], Jakob Manthey[2,3,4], Christiane Sybille Schmidt[2], Marina Piazza[5‡], Ines Bustamante Chavez[5‡], Guillermina Natera[6‡], Natalia Bautista Aguilar[6‡], Graciela Yazmín Sánchez Hernández[6‡], Juliana Mejía-Trujillo[7‡], Augusto Pérez-Gómez[7‡], Antoni Gual[8,9,10‡], Hein de Vries[11‡], Adriana Solovei[11‡], Dasa Kokole[11‡], Eileen Kaner[1‡], Carolin Kilian[3‡], Jurgen Rehm[2,3,12,13,14,15‡], Peter Anderson[1,11‡], Eva Jané-Llopis[11,12,16‡]

1 Population Health Sciences Institute, Newcastle University, Newcastle Upon Tyne, United Kingdom, 2 Center for Interdisciplinary Addiction Research (ZIS), Department of Psychiatry and Psychotherapy, University Medical Center Hamburg-Eppendorf, Hamburg, Germany, 3 Institute for Clinical Psychology and Psychotherapy, TU Dresden, Dresden, Germany, 4 Department of Psychiatry, Medical Faculty, University of Leipzig, Leipzig, Germany, 5 Mental Health, Alcohol, and Drug Research Unit, School of Public Health and Administration, Universidad Peruana Cayetano Heredia, Lima, Peru, 6 Instituto Nacional de Psiquiatría Ramón de la Fuente Muñiz, CDMX, Mexico, 7 Corporación Nuevos Rumbos, Bogotá, Colombia, 8 Addictions Unit, Psychiatry Dept, Hospital Clínic, Barcelona, Spain, 9 Red de Trastornos Adictivos, Instituto Carlos III, Madrid, Spain, 10 Institut d'Investigacions Biomèdiques August Pi Sunyer (IDIBAPS), Barcelona, Spain, 11 Department of Health Promotion, CAPHRI Care and Public Health Research Institute, Maastricht University, Maastricht, The Netherlands, 12 Institute for Mental Health Policy Research, CAMH, Toronto, Ontario, Canada, 13 Dalla Lana School of Public Health, University of Toronto, Toronto, Ontario, Canada, 14 Department of Psychiatry, University of Toronto, Toronto, Ontario, Canada, 15 Department of International Health Projects, Institute for Leadership and Health Management, I.M. Sechenov First Moscow State Medical University, Moscow, Russian Federation, 16 Univ. Ramon Llull, ESADE, Barcelona, Spain

☯ These authors contributed equally to this work.
‡ MP, IBC, GN, NBA, GYSH, JMT, APG, AG, HV, AS, DK, EK, CK, JR also contributed equally to this work. PA and EJL are joint senior authors on this work.
* amy.odonnell@newcastle.ac.uk

## Abstract

### Introduction

Implementation of evidence-based care for heavy drinking and depression remains low in global health systems. We tested the impact of providing community support, training, and clinical packages of varied intensity on depression screening and management for heavy drinking patients in Latin American primary healthcare.

### Materials and methods

Quasi-experimental study involving 58 primary healthcare units in Colombia, Mexico and Peru randomized to receive: (1) usual care (control); (2) training using a brief clinical package; (3) community support plus training using a brief clinical package; (4) community

**Data Availability Statement:** All relevant data are within the manuscript and its Supporting information files.

**Funding:** The research leading to these results or outcomes has received funding from the European Horizon 2020 Programme for research, technological development and demonstration under Grant Agreement no. 778048-Scale-up of Prevention and Management of Alcohol Use Disorders and Comorbid Depression in Latin America (SCALA). Additionally, we thank the financing granted by the Rio Arronte Foundation to enable the implementation of our study in Mexico. Participant organisations in SCALA can be seen at: www.scalaproject.eu. The views expressed here reflect those of the authors only and the European Union is not liable for any use that may be made of the information contained therein. The Funder was not involved in the study design. The funder had no role in study design, data collection and analysis, decision to publish, or preparation of the manuscript.

**Competing interests:** The authors have declared that no competing interests exist.

**Abbreviations:** AUD, Alcohol Use Disorder; AUDIT, Alcohol Use Disorder Identification Test; AUDIT-C, Alcohol Use Disorder Identification Test-Consumption; LMIC, Low and Middle Income Countries; PHC, Primary Health Care; PHQ-2, Patient Health Questionnaire; SCALA, Scaling up the Prevention and Management of Alcohol Use Disorders and Comorbid Depression in Latin America.

support plus training using a standard clinical package. Outcomes were proportion of: (1) heavy drinking patients screened for depression; (2) screen-positive patients receiving appropriate support; (3) all consulting patients screened for depression, irrespective of drinking status.

## Results

550/615 identified heavy drinkers were screened for depression (89.4%). 147/230 patients screening positive for depression received appropriate support (64%). Amongst identified heavy drinkers, adjusting for country, sex, age and provider profession, provision of community support and training had no impact on depression activity rates. Intensity of clinical package also did not affect delivery rates, with comparable performance for brief and standard versions. However, amongst all consulting patients, training providers resulted in significantly higher rates of alcohol measurement and in turn higher depression screening rates; 2.7 times higher compared to those not trained.

## Conclusions

Training using a brief clinical package increased depression screening rates in Latin American primary healthcare. It is not possible to determine the effectiveness of community support on depression activity rates due to the impact of COVID-19.

## Introduction

Depression is the most common mental health disorder, affecting more than 300 million people worldwide [1]. A top three leading cause of disability and a relevant factor for excess all-cause mortality [2–4], depressive disorders also lead to substantial negative social and economic impacts [5–7]. There is a strong reciprocal relationship between alcohol use disorders (AUD) and major depression; heavy drinking increases the risk for developing a depressive disorder [8, 9] and there is a high prevalence of alcohol use disorders amongst those diagnosed with depression [10, 11]. Whilst available data suggest higher rates of depressive disorders are experienced in populations living in wealthier regions [4], depression poses a substantial public health challenge for low-and-middle income countries (LMIC), in part due to less developed health systems and limited resources to support those with mental ill-health [12, 13].

There is substantial evidence of the effectiveness of preventative approaches to reduce the health burden related to depression [14, 15], including in LMIC populations [16]. However, translation of these interventions into routine practice remains low in global health systems [17]. Despite a lifetime prevalence of 12.5% and a 12-month prevalence of 6.7% for major depression across Latin America, only around 40% of those affected receive appropriate care [18], with even higher treatment gaps found in specific population subgroups. In Peru and Mexico, for example, up to 95.6% of affected older adults have never received any support for their depression [19–21]. Barriers to delivery in this region include a range of systemic, organisational, and individual level factors [6], with low numbers of mental health specialists and limited resources for mental healthcare, highlighted as particular concerns [13, 22, 23].

Given that the majority of individuals with depressive symptoms are first seen in primary health care (PHC) [24], and based on the aforementioned high comorbidity with AUD [25],

supporting PHC providers to more efficiently identify and support heavy drinking patients with depression could help address the mental health treatment gap in LMICs [16, 26, 27]. Few studies of combined psychological interventions for co-morbid AUD and depression exist but results to date have been promising [28, 29]. In PHC, the brief two-question Patient Health Questionnaire (PHQ-2) is generally recommended as an initial depression screening tool in most clinical guidelines [16, 30, 31], with reasonable levels of sensitivity and specificity reported [32, 33]. For patients identified as in need of support, a range of evidence-based, brief, psychological interventions and pharmacological treatments are available that are feasible, effective and acceptable for delivery in primary care [29, 34–37].

At the same time, a review of evidence in this field showed that LMIC PHC providers have raised concerns about their capacity to manage complex mental health issues given resource and expertise constraints, especially where social risk factors are concerned [23]. As such, additional support, including provision of training and manualised guidance as part of a wider, integrated, collaborative care model (CCM), is likely to be needed to enable PHC providers to effectively implement these interventions in routine practice [27]. Collaborative care is based on principles of the chronic care model [38, 39] and involves integrating behavioural health interventions within PHC, typically using a multi-professional approach to patient care, structured management, scheduled patient follow-ups, and enhanced inter-professional communication [40]. A substantial body of literature supports the clinical- and cost-effectiveness of using CCM to improve identification treatment of depression and other mental health conditions in PHC [41–43], including more limited evidence for its application in the management of substance use disorders including alcohol [44]. However, most existing research in this field has been conducted in high income countries, predominantly the USA [45, 46].

In response to this knowledge gap, the international SCALA project (Scale-up of Prevention and Management of Alcohol Use Disorders and Comorbid Depression in Latin America, www.scalaproject.eu) seeks to test the impact of a range of strategies including provision of community action and support with a clinical and training package of varied intensities on rates of identification and support for heavy drinking patients with depression in PHC in Colombia, Mexico, and Peru [47]. We hypothesised that:

**Hypothesis 1 (H1)**: the presence of community action and support would lead to improved rates of depression screening and management;

**Hypothesis 2 (H2)**: the provision of training would lead to improved rates of depression screening and management; and

**Hypotheses 3 (H3)**: in the presence of community action and support, use of a less intensive (brief tailored) clinical package and training would not lead to worse rates of depression screening and management compared to those using the standard version.

In this paper, we present results of the first interim evaluation (planned for month six but brought forward to month five due to the impact of the COVID-19 pandemic) for our secondary outcomes: 1) proportion of patients identified as heavy drinkers who are also screened for depression; 2) proportion of patients screened positive for depression receiving advice and/or treatment for comorbid depression; and 3) proportion of all consulting patients screened for depression, irrespective of drinking status. Results for our primary outcomes, rates of measurement of alcohol consumption of PHC patients and proportion of heavy drinking patients receiving support, are reported elsewhere [48].

## Materials and methods

### Design

A quasi-experiment to compare the impact of a tailored clinical and training package of varied intensity plus community action and support measures on changes in measurement of alcohol consumption and identification of depression between PHC units (PHCUs) in intervention and control municipal areas in the cities of Bogotá (Colombia), Mexico City (Mexico) and Callao–Lima (Peru) (see Fig 1) [47]. Components of the clinical and training package and community action and support measures were based on existing empirical evidence of effective

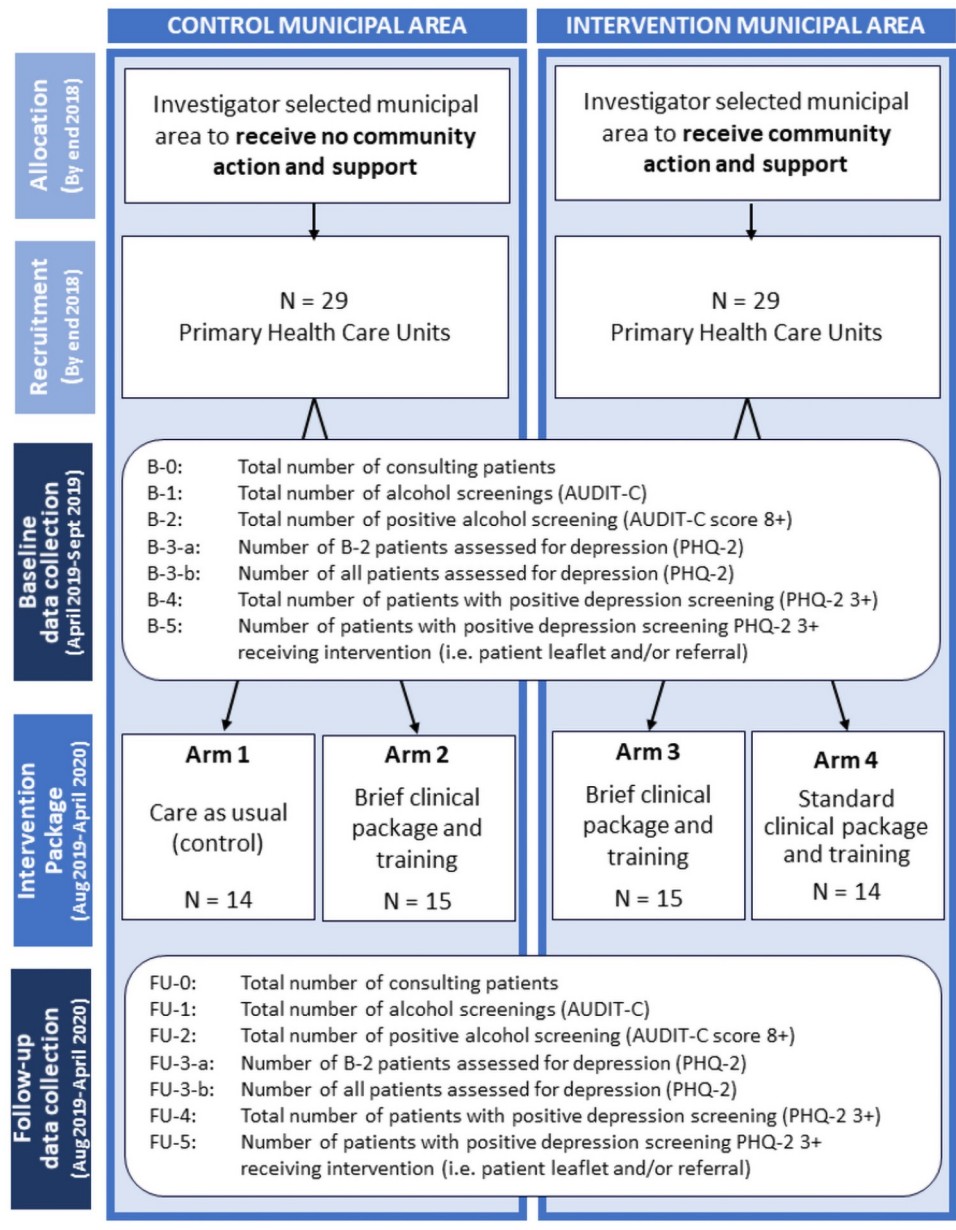

**Fig 1. SCALA study flowchart.**

implementation strategies to promote delivery of PHC-based support for AUD and depression [49–54].

Intervention and control municipal areas were investigator-selected within each city based on comparability in terms of socio-economic characteristics, and with sufficient geographical separation to minimize spill-over effects. In Bogota, the intervention municipal area was Soacha (population [pop]: 93,154) and control comprised Funza (pop: 112.254) and Madrid (pop: 93,154). In Mexico City, the intervention municipality area comprised Tllapan (pop: 650.567), Benito Juárez (pop: 385,439) and Álvaro Obregón (pop: 727,034); control Miguel Hidalgo (pop: 372,889) and Xochimilco (pop: 415,007). In Lima, the intervention area was in Callao (pop: 451,260), with control comprised of Chorillos (pop: 314,241) and Santiago de Surco (pop: 329,152). Within these six municipal areas (one intervention and one control area per city), a total of 58 PHCUs were recruited; 29 in the intervention areas and 29 in the control areas, representing 622 eligible providers. Eligible providers were defined as any PHC provider involved in medical and/or preventive care (such as, but not limited to, doctors, nurses, psychologists, nurse technicians, social workers or midwifes) and providing written informed consent for their participation.

Within the control municipal areas, 14 PHCUs were randomly allocated to control (Arm 1) and 15 to receive a brief clinical intervention and provider training package alone (Arm 2). Within the intervention municipal areas, all 29 PHCUs received community action and support. Of these, 15 units were randomly allocated to receive a brief (less intensive) clinical intervention and provider training package (Arm 3) and 14 to receive a standard (more intensive) clinical intervention and provider training package (Arm 4). Random allocation of PHCUs was stratified by country and undertaken using an Excel random number generator.

## Intervention package

The clinical package comprised: 1) alcohol measurement and depression screening instruments (three question Alcohol Use Disorder Identification Test-Consumption (AUDIT-C); two question Patient Health Questionnaire (PHQ-2) [33, 55]); 2) patient advice material for those identified with heavy drinking and/or depression; and 3) provider guidelines. Two versions of the package were developed: a standard, more intensive version, essentially that used in common clinical practice (Arm 4); and a brief, simplified version, designed to be deliverable in a shorter time (Arms 2 and 3). All clinical materials were tailored for local use based on the Tailored Implementation in Chronic Disease (TICD) checklist [56].

The training package (Arms 2–4) employed a modelling methodology [57, 58], using videos and role-play, with content tailored to the PHC settings and health systems in each country. As with the clinical package, brief (Arms 2 and 3) and standard (Arms 4) versions were developed (fully described elsewhere [47, 59]). Training sessions lasted between two and four hours and were delivered by existing country-based trainers who had previously attended a SCALA-specific train-the trainers-programme. Over 70 percent of providers attended at least one training session (72.3%), with a higher rate reported for Arms 2 and 4 (74.1% and 76.9% respectively) compared to Arm 3 (66.3%) [see 48]. During the implementation period, booster training sessions were offered, which included summary reminders of key concepts and the care pathway process alongside tailored 'troubleshooting' based on providers' experience to date. However delivery varied due to the impact of the COVID-19 pandemic; Colombia implemented all planned booster sessions but there was only partial delivery in Peru (13 out of 15 sessions) and Mexico (seven out of 14 sessions).

The community action and support measures (Arms 3 and 4) were based on the Institute for Healthcare Improvement Going to Scale Framework [60], and consisted of five core blocks

of activity: 1) creation of local stakeholder groups to advise on the tailoring of materials, support implementation and review drivers of successful action; 2) appointment of local project champions to advocate for successful implementation across their municipal area; 3) evidence-based adoption mechanisms including local promotion of the advantages of and need for SCALA alongside PHCU-level audit and feedback measures; 4) evidence-based support systems including the creation of an information exchange and learning system for participating PHC providers; and 5) community-based communication campaigns. Further details of the community action and support measures are described in Solovei et al. [61]. Due to the COVID-19 pandemic, however, only limited implementation of these measures was possible. In particular, minimal progress had been made with the municipal communication campaigns planned for delivery in each country.

## Clinical procedure

A clinical pathway specified how alcohol and depression screening, intervention delivery and/or referral procedures should be implemented in routine consultations (see Fig 2). PHC providers (all Arms) were asked to measure the alcohol consumption of all adult patients who consulted for any reason using the three question AUDIT-C [62]. Patients who scored 8+ with AUDIT-C were also to be screened for depression using the PHQ-2, and managed as appropriate, as well as being advised to reduce their alcohol consumption unless there were clinical indications for referral. As explained elsewhere, although the standard AUDIT-C cut-off scores are usually set at five for both men and women, or five for men and four for women [63], in SCALA we raised this to 8+ to enhance clinical credibility and ensure limited capacity for advice-giving was targeted at those patients in most need [47]. PHC providers (all Arms) were asked to provide those scoring PHQ-2 3+ with a patient information leaflet about depression, and/or referral to further treatment, as appropriate. Arm 4 differed from Arms 1–3 in having a lengthier assessment and advice-giving session. PHC providers used paper tally sheets to document: patient screening scores; consultation outcome (advice given, leaflet provided, patient referred); and key socio-demographic characteristics. Patients who had previously completed AUDIT-C were excluded from completion a second time.

## Data collection

Data were collected between April 2019 and April 2020. Before the start of baseline data collection, we obtained key characteristics of the participating PHC units, including the number of providers working in the unit by profession (Doctor, Nurse, Psychologist, Nurse Technician, Social worker, Midwife, Other), and the number of registered adult patients (see S1 Table). From baseline thereon, providers reported the number of adult consultations conducted per month. As data collection was staggered across countries and PHCUs (for further details, see below), only six months of data (one-month baseline and five months implementation) were completed in all PHC units during the first 12 months of implementation.

## Definition of variables for analyses

We present the variables used for statistical analyses separately for: A) exposure; B) outcome; and C) adjustment variables.

 A). **Exposure variables**. *Community support (H1)*: Arm 3 (with community support) versus Arm 2 (without community support)
 *Training (H2)*: Arm 2 (with training) versus Arm 1 (without training)

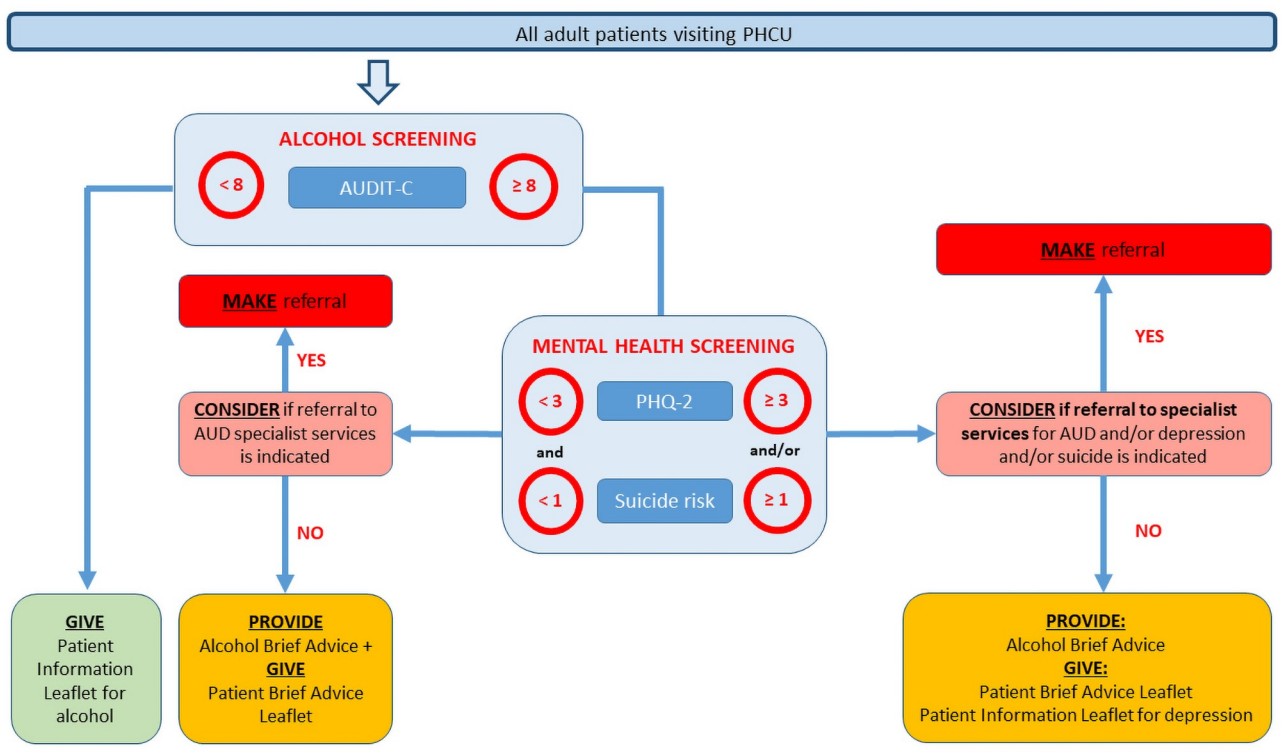

**Fig 2. SCALA clinical pathway.**

*Intensity of clinical intervention package (H3)*: Arm 4 (standard more intensive package) versus Arm 3 (brief less intensive package)

B). **Outcome variables**. *Outcome 1*: The cumulative share of heavy drinking patients also assessed for depression (preregistered outcome definition).
This outcome was calculated as the number of consulting adults with an AUDIT-C score of 8+ who completed the PHQ-2 (B-3a/FU-3a) divided by the total number of patients with an AUDIT-C score of 8+ per participating provider (B-2/FU-2). This was calculated for those providers identifying at least one patient as a heavy drinker during the study period (N = 129).
*Outcome 2*: The cumulative share of patients at-risk for depression receiving appropriate interventions (preregistered outcome definition)
This outcome was calculated as the number of adults with a PHQ-2 score of 3+ who received a patient leaflet and/or referral for their depression (B-5/FU-5) divided by the total number of patients with a PHQ-2 score of 3+ per participating provider (B-4/FU-4). This was calculated for those providers identifying at least one single patient as being at-risk for depression during the study period (N = 85).
*Outcome 3*: The cumulative coverage of depression screenings per 1,000 consulting patients (new outcome definition).
This outcome was calculated as the number of consulting adults who completed the PHQ-2 (B-3b/FU-3b) divided by the total number of consulting patients per participating provider (B-0/FU-0) that is the rate of screening in the general consulting patient population, not just those scoring 8+ on the AUDIT-C. This was calculated for all providers participating in the study (N = 606) and expressed as number of depression screens per 1,000

consulting patients. In addition to predefined outcomes 1 and 2, this would allow us to understand the extent to which SCALA had led to wider impacts on depression identification and management practices in the participating PHC units.

C). **Adjustment variables**. Models were adjusted for 1) provider-level characteristics (gender, age, being a medical doctor vs other professions) and 2) dummy-coded country variables, to account for differences between countries. Adjustment for baseline levels would have excluded providers who joined the study after baseline (N = 98) and was thus not carried out.

## Sensitivity analyses for hypothesis 2

Due to the limited training coverage highlighted earlier, in addition to the intention-to-treat analyses (main analyses), we performed per-protocol analyses to compare providers who attended the training session in Arm 2 with those who had not received any training in Arms 1 and 2. As with the main analyses, all sensitivity analyses were adjusted for variations across country, sex, age, and profession.

## Statistical analyses

Analyses were conducted at the provider level, with n = 16 out of N = 622 providers from Colombia removed as they worked in different arms (employed by multiple PHCUs) during data collection, leaving a total sample of N = 606 providers.

As outcome 1 and 2 were proportions, fractional response regressions were performed to test the hypotheses. For outcome 3 (rate of depression screenings per 1,000 consulting patients), the distribution of the variable was assessed before performing statistical analyses. As the distribution of outcome 3 could best be described with a negative binomial distribution (shown in S1 Fig), negative binomial regression analyses were performed to test hypotheses for this outcome. Further, all analyses for outcome 3 were weighted for the number of consultations, allowing interpretation of the findings relative to the number of consultations. The weight variable was normalized by dividing the number of consultations (denominator of outcome 3) by the mean number of consultations across all providers. All three dependent variables were compared by exposure variable, i.e. presence of community support (H1), training provision (H2), and intensity of clinical intervention package (H3).

In all analyses, country differences, gender, age, and profession type were accounted for. For each outcome, we performed additional models testing for interaction of country and exposure variables. Findings were reported if they were different to the main models and likelihood ratio tests indicated improved model fit. Additional analyses were also conducted to identify possible PHCU cluster effects in the data. As results suggested that cluster effects were negligible, all analyses were conducted without accounting for clustering (see S1 Fig).

Following the protocol deviation to analyse data collected during the first five months of implementation, all Peruvian PHCUs completed their fifth month by early February 2020. In Colombia and Mexico, the implementation period was staggered, resulting in several PHCUs completing their fifth month of data collection only post-data closure. For these PHCUs, the last reported cumulative coverage rate was carried forward, assuming no further measurements (see S3 Table). The missing data due to incomplete data collection in select PHCUs was estimated to be of negligible extent (see S3 Table).

All analyses were run with R version 4.0.2 [64].

### Ethical approval and consent to participate

The Ethics Committee of the Technical University of Dresden granted the study final ethical approval on 12 April 2019 (Ref: EK90032018). All participating PHCUs and PHC providers provided signed informed consent.

## Results

### Sample characteristics

Of the N = 606 providers participating in the study between April 2019 and March 2020, N = 508 providers joined during the baseline assessment period, with the remainder joining the study post-baseline. Of these, N = 363 completed both baseline and the subsequent five months of the implementation period. Between baseline assessment and completion of the five months implementation period, N = 180 of the N = 606 providers dropped out of the study. Across the four study Arms, the distribution of females (Chi$^2$ test: p = .098), doctors (Chi$^2$ test: p = .200), dropouts (Chi$^2$ test: p = .275), training rate (Chi$^2$ test: p = .201) did not differ significantly. However, providers in Arm 4 were younger than those in Arm 2 (ANOVA: p = .045; post-hoc test: p = .003). Across all study Arms, participating providers included: 40.2% Doctors; 13.8% Nurses; 7.9% Psychologists; and 37.8% dother professions (e.g. social workers, midwives, dentists, and nutritionists). See Table 1 for full details.

### Outcome 1: Share of heavy drinking patients screened for depression

N = 129 providers identified N = 615 heavy drinking patients (the remaining n = 477 providers had no patients measured with an AUDIT-C score of 8+ during this period), out of which N = 550 (89.4%) were screened for depression. As reported in Table 2, we observed similar mean screening rates across all four study Arms.

As illustrated in Table 3 (for full regression results including country effects, see S1–S3 Tables), we did not find any evidence that community support (H1) or training (H2) had a significant effect on the rate of depression screening amongst heavy drinking patients. Fewer heavy drinking patients were screened for depression by providers working with the standard compared to the brief package (H3), but the difference was not significant (p = 0.071). For H3, additional models suggested no overall effect of the package length on delivery rates. This result was in line with hypotheses 3, postulating no increased screening activities in Arm 4. In

**Table 1. Characteristics of participating providers.**

|  | Arm 1 | Arm 2 | Arm 3 | Arm 4 |
|---|---|---|---|---|
| *Sample characteristics* | | | | |
| N (PHC units) | 14 | 15 | 15 | 14 |
| N (providers) | 118 | 194 | 160 | 134 |
| % Female | 78.0% | 75.8% | 77.5% | 66.4% |
| Age (SD) | 38.8 (12.2) | 41.4 (12.5) | 37.8 (12.5) | 37.3 (11.7) |
| % Doctor | 37.3% | 37.6% | 41.3% | 48.5% |
| Mean no. of months providers participated in study, max = 6 (SD) | 5.0 (1.7) | 4.5 (2.1) | 4.5 (2.0) | 4.8 (1.9) |
| Proportion of providers dropped out during study period | 33.9% | 24.7% | 32.5% | 29.9% |
| Proportion of providers attending at least one training session | N/A | 74.1% | 66.3% | 76.9% |

Note. SD = Standard Deviation.

**Table 2. Descriptive results for the three outcome measures by arm.**

| | Arm 1 | Arm 2 | Arm 3 | Arm 4 |
|---|---|---|---|---|
| *Outcome 1: Cumulative share of heavy drinking patients assessed for depression* | | | | |
| N (providers with at least one hazardous drinker identified) | 11 | 44 | 39 | 35 |
| Nominator (cumulative no. of depression screens): mean (SD) | 2.5 (2.5) | 4.8 (5.3) | 4.9 (6.5) | 3.5 (4.7) |
| Denominator (cumulative no. of hazardous drinkers identified): mean (SD) | 2.6 (2.5) | 5.3 (5.3) | 5.3 (7.2) | 4.2 (5.6) |
| Outcome 1: mean % | 90.90% | 89.50% | 93.50% | 83.80% |
| *Outcome 2: Cumulative share of patients at-risk for depression receiving appropriate interventions* | | | | |
| N (providers with at least one patient at risk identified) | 6 | 28 | 25 | 26 |
| Nominator (cumulative no. of at-risk patients receiving appropriate interventions): mean (SD) | 1.8 (1.7) | 1.9 (2.8) | 1.7 (3.0) | 1.5 (2.6) |
| Denominator (cumulative no. of at-risk patients): mean (SD) | 2.2 (1.6) | 2.5 (2.7) | 3.0 (3.4) | 2.8 (2.8) |
| Outcome 2: mean % | 77.80% | 69.00% | 35.00% | 46.70% |
| *Outcome 3: Cumulative rate of depression screens per 1,000 consulting patients* | | | | |
| N (all providers) | 118 | 194 | 160 | 134 |
| Nominator: cumulative no. of depression screens | 40 | 407 | 508 | 421 |
| Denominator: cumulative number of consulting patients | 181,466 | 489,323 | 1,019,903 | 597,332 |
| Outcome 3: mean rate (SD) | 0.2 (1.5) | 0.8 (2.3) | 0.5 (2.3) | 0.7 (1.5) |

Note. SD = Standard Deviation.

additional analyses, models testing for interaction effects showed no improved data fit. Thus, for outcome 1, we find no support for hypotheses 1 and 2 but accept hypothesis 3.

## Outcome 2: Share of population at-risk for depression receiving appropriate interventions

N = 85 of the 129 providers identified 230 patients with a PHQ-2 score of 3+ (the remaining N = 44 providers identified no patients with a PHQ score of 3+) (see Table 2). Of the 230 eligible patients, N = 147 received appropriate interventions (63.9% of those patients who screened positive for depression), which were either referred to a specialist (N = 10), provided with a leaflet (N = 107), or both (N = 30). As reported in Table 2, we observed higher intervention

**Table 3. Results of regression analyses for evaluating the three hypotheses for three outcomes.**

| Reporting exponentiated coefficients for exposure variable[a] | Hypothesis 1 (Arm3>Arm2) | Hypothesis 2 (Arm 2>Arm 1) | Hypothesis 3 (Arm 4>Arm 3) |
|---|---|---|---|
| Outcome 1: % of heavy drinking drinkers screened for depression[b] | 1.75 (0.58 to 5.64) | 3.04 (0.12 to 76.86) | 0.36 (0.11 to 1.04) |
| Outcome 2: % of patients at-risk for depression intervened[c] | 0.25 (0.06 to 0.95) | 1.15 (0.07 to 14.89) | 0.83 (0.21 to 3.19) |
| Outcome 3: Rate of depression screens per 1,000 consulting patients[d] | 0.90 (0.63 to 1.29) | 3.52 * (1.70 to 7.82) | 1.27 (0.89 to 1.80) |

Note: For outcome 1 and 2, exponentiated coefficients of fractional response regression analyses are presented, which should be interpreted as percentage increase associated with one unit increase in predictor variable. For outcome 3, exponentiated coefficients of negative binomial regression analyses, which should be interpreted as Incidence Rate Ratios. Numbers in brackets denote 95% confidence intervals.

[a] Exposure variable defined by hypothesis: H1: without (base) vs with municipal support, H2: without (base) vs with training, H3: short (base) vs standard package

[b] Fractional response regression analyses with n = 83 providers for H1, n = 55 providers for H2 and n = 74 providers for H3, controlling for country differences, gender, age and profession.

[c] Fractional response regression analyses with n = 53 providers for H1, n = 34 providers for H2 and n = 51 providers for H3, controlling for country differences, gender, age and profession.

[d] Negative binomial regression analyses with n = 349 providers for H1, n = 309 providers for H2 and n = 287 providers for H3, controlling for country differences, gender, age and profession.

*p<0.01.

rates in Arms 1 and 2 compared to Arms 3 and 4. The number of screen-positive patients also varied substantially across Arms (Arm 1: 13; Arm 2: 70; Arm 3: 74; Arm 4: 73).

We did not find any evidence that community support (H1), training (H2) or intensity of the package (H3), improved the likelihood of receiving interventions among patients screening positive for depression (see Table 3). Alternatively, we found community support to be associated with lower rates of receipt of appropriate interventions amongst patients screening positive for depression. Inclusion of interaction terms did not alter the presented findings. Thus, for outcome 2, we find no support for hypotheses 1 and 2 but accept hypothesis 3.

### Outcome 3: Rate of depression screens per 1,000 consulting patients

N = 606 providers had N = 2,288,023 consulting patients, out of which N = 1,376 were screened for depression, a rate of 1 per 1,663 consulting patients (**see** Table 2). Whilst only heavy drinking patients (AUDIT-C score of 8+) were eligible for depression screening according to the clinical pathway, only 550 out of 1,376 patients screened for depression scored 8+ on the AUDIT-C. As shown in Table 2, the number of monthly consultations varied considerably across arms (Arm 1: 280; Arm 2: 836; Arm 3: 1217; Arm 4: 809). Similarly, we observed higher depression screening rates in Arms 2, 3 and 4 as compared to Arm 1.

As illustrated in Table 3, we did not find any evidence that community support (H1) or intensity of the package (H3) had a significant effect on the coverage of depression screening amongst consulting patients. However, a significantly higher rate of consulting patients was screened for depression by providers assigned to the training condition (H2). Findings from additional models confirmed the overall training effect on depression screening rates, but that this varied by country (see S4 Table). Thus, for outcome 3, we find no support for hypothesis 1 but accept hypotheses 2 and 3.

### Sensitivity analyses

In sensitivity analyses testing H2 (effects of training on the three outcomes, see S5 Table), we compared N = 120 out of N = 194 providers in Arm 2 who attended at least one training session to the remaining N = 192 providers in Arms 1 and 2 who did not receive any training. Sensitivity analyses confirmed no effect of training with regards to both depression screening activities for heavy drinking patients (outcome 1; exponentiated coefficient: 5.0, 95% CI: 0.6 to 75.2, $p = 0.172$) and with regards to the likelihood of patients at-risk for depression receiving appropriate interventions (outcome 2; exponentiated coefficient: 0.6, 95% CI: 0.05 to 4.8, $p = 0.608$). Sensitivity analyses also confirmed the training effect on overall rates of depression screening, with a very similar effect size but narrower confidence interval compared to the main analyses (outcome 3; incident rate ratio: 3.7, 95% CI: 2.1 to 6.7, $p < 0.001$). These results indicate that the depression screening rate amongst trained providers was 3.7 times higher compared to those not trained. Limiting analyses to providers in Arm 2 (i.e. comparing those that attended (N = 120) with those that did not attend the training (N = 74)) attenuated the effect strength but the difference remained significant (incident rate ratio: 2.7, 95% CI: 1.3 to 5.9, $p = 0.007$).

### Discussion

We tested the impact of community support, training, and provision of a tailored clinical intervention package of varied intensity on rates of screening and management of depression in heavy drinking primary care patients in Colombia, Mexico and Peru. We found that 89.4% of patients identified as heavy drinkers (AUDIT-C score 8+) were also assessed for depression; this proportion was unaffected by provision of either training or municipal support. Of those

patients who screened positive for depression, 63.9% received some type of intervention, with the proportion unaffected by either training or intensity of the clinical package, but lower amongst the community support arm. However, although we found that the overall proportion of consulting patients assessed for depression (i.e. irrespective of drinking status) was unaffected by either the presence of community support or the intensity of the clinical package, rates were higher amongst those providers who had received training. These latter findings are consistent with our five-month alcohol-related outcome results, where we found that participating in training increased the proportion of patients whose alcohol consumption was measured [48].

With more alcohol measurements taking place overall, and heavy use being a trigger for depression assessment according to the SCALA clinical pathway, it is not unexpected that training would lead to a higher proportion of consulting patients being assessed for depression. Thus, although training had no direct impact on the proportion of patients with an AUDIT-C score of 8+ assessed for depression, we found an indirect impact of training on the overall proportion of patients assessed for depression. Moreover, as we found non-superiority of the standard package, we conclude that the shorter less intense package can be implemented as the norm [48].

There has been limited implementation of appropriate care in PHC for patients with heavy drinking and depressive disorders, particularly in LMICs [18, 19]. Evidence from systematic literature reviews suggests that multifaceted interventions that promote collaborative models of care can improve the identification and management of depression in PHC, at least in the short-term [65]. Key components of successful interventions identified by previous studies include clinician education, nurse-led case management, structured patient monitoring processes, and health-system action to increase integration between primary and specialist care, via a standardised clinical pathway; some of which strategies were tested in the study presented here [54]. As already highlighted however, there have been limited studies of these approaches in LMIC settings [45, 46]. As such, when scaling-up practice outside high income regions, others have highlighted the need for tailored models of care and associated improvement strategies that take into account relevant contextual factors [46], including local conditions of mental health systems [66, 67].

On this basis, we had anticipated that the implementation of community actions to provide a more supportive local delivery context would lead to more patients being assessed for heavy drinking, and thus in turn depression. To date, we have not demonstrated this, potentially due to the negative impact of COVID-19 on the introduction of associated local actions and measures; further compounded by the fact that their effects are likely to accumulate over time (the full implementation period for SCALA was 18 months). For example, the municipal communication campaigns were a critical component of the community actions planned in each country. In Mexico and Peru, however, the communication campaigns had only been implemented for three to four weeks prior to the introduction of COVID-19 related social distancing measures and lockdowns and had not yet started in Colombia. Thus, it is premature to conclude that community support does not lead to a higher proportion of patients assessed for depression.

Several implications for policy, practice, and subsequent phases of research emerge from our results. Our findings suggest that it is feasible for PHC providers to assess patients for depression using PHQ-2 in the presence of heavy drinking as indicated by an AUDIT-C score of 8+. However, it is important to stress that because we used a higher than normal cut-off score for AUDIT-C, very few heavy drinkers were identified during this initial implementation period. This meant that whilst almost nine out of ten identified heavy drinkers were screened for depression, the overall proportion of patients assessed for depression was very low (1 in

1,663 consulting adult patients), although this rate did increase with training provision. More-over, although the majority (63.9%) of heavy drinking patients that also screened positive for depression were recorded as receiving some type of intervention or treatment, 83 eligible patients did not appear to receive any support. At present, we can only speculate on the possible reasons for lack of recorded care; forthcoming findings from the planned process evaluation may provide greater insight on this issue [59]. On balance, we suggest that to increase the coverage of depression assessment and relevant care provision in adult patient populations, PHC providers require tailored skills-based training. The results presented here suggest that this training can be of relative short duration (two hours).

We plan to restart implementation as lockdowns and social distancing measures are relaxed in Colombia, Mexico, and Peru. We will review and refocus the community support measures, telescoping a sustained and intense action within a short period of time during a second six-month implementation phase. We will also adopt telemedicine approaches and introduce digital applications for implementing the clinical package where possible, including the provision of web-based training for PHC providers. Recent research from Colombia indicates that tablet-based depression and alcohol screening and clinical guidance can be successfully implemented in PHC settings, leading to increased rates of diagnoses, particularly for depressive disorders [68]. Similarly, an additional study in Peru showed that the use of screening app to identify at-risk patients followed by a motivational and reminder short message service (SMS) was feasible for implementation in PHC settings [69].

Finally, given the adverse impact that COVID-19 and associate social-distancing and lockdown measures have had on mental health, with increased rates of depression reported in Latin America and elsewhere [70], we plan to assess all presenting patients for depression with PHQ-2, rather than limiting assessment to those with a high AUDIT-C score. This is likely to necessitate a change in focus for planned the communication campaigns, to raise awareness of the support available in PHC in the intervention municipal areas [71].

## Strengths and limitations

To our knowledge this is the first multi-country study testing the implementation of interventions for comorbid heavy drinking and depression in middle-income countries. The scale-up strategy and tailoring process were guided by established models and frameworks, namely the Institute for Health Care Improvement [60] and the Tailored Implementation in Chronic Disease (TICD) checklist [56].

The study has three main weaknesses. First, due to the impact of COVID-19-related social distancing and lockdown measures, we decided to bring the six-month interim evaluation forward to month five. Most data collection was completed by month five (March 2020), immediately before COVID-19-related travel restrictions were introduced, and the work of the PHCUs in the countries was realigned. However, the shortened timeframe means that results are likely to be more conservative as effects would be expected to increase over time. More importantly, as we were unable to fully implement the planned community action and support activities, we are unable to draw conclusions with regards to their impact. Second, we were not able to randomly allocate the participating municipal areas in each country. However, we did allocate randomly PHCUs to arms within the municipal areas, and baseline data shows that providers in both areas were similar in key socio-demographic variables. We did not have detailed information on their previous skills and expertise in the field of alcohol and/or mental health. However, some of the Mexican providers had previously received on mental health topics, including depression, through the WHO Mental Health Gap Action Programme (mhGAP, see [72]); and in all countries, participating psychologists tended to be more familiar with

managing mental health issues, including depression, compared to other generalist primary care clinicians. Third, and finally, the low rates of patients identified as heavy drinkers during this initial phase of the SCALA study in turn limited the potential coverage of depression assessments in consulting patients (as according to our clinical pathway, only patients who scored 8+ on AUDIT-C should be subsequently screened for depression). Looking forward, by assessing all presenting patients for depression with PHQ-2, rather than limiting assessment to those with a high AUDIT-C score, we hope to both respond to local emerging mental health needs and boost activity rates in this area.

## Conclusions

Our five-month interim analysis indicates that training PHC providers to identify and support heavy drinking patients with depression, increases rates of depression screening for both those drinking excessively and adult patients in general. Due to the impact of COVID-19 restrictions on the implementation of this study, it is not possible to determine the effectiveness of community support on depression activity rates at this stage. We are also limited in our assessment of whether providing a less versus more intensive clinical intervention package encourages higher rates of screening and advice delivery by PHC providers. Going forward, for those countries experiencing prolonged social distancing and lockdown measures due to COVID-19, tele-health and digital technology could provide useful options to support PHC-based screening and care for heavy drinking patients with depression.

## Supporting information

**S1 Fig. Density of outcome 3 cumulative coverage by Arm.** The red and yellow lines indicate hypothetical negative binomial and poisson distributions, respectively, generated from parameters of the empirical arm-specific distribution.
(TIF)

**S1 Table. Results of regression analyses for evaluating hypotheses 1–3 for outcome 1 (cumulative share of heavy drinking patients assessed for depression).**
(DOCX)

**S2 Table. Results of fractional response regression analyses for evaluating hypotheses 1–3 for outcome 2 (cumulative share of patients at-risk for depression receiving appropriate interventions).**
(DOCX)

**S3 Table. Results of negative binomial regression analyses for evaluating hypotheses 1–3 for outcome 3 (cumulative rate of depression screens per 1,000 consulting patients).**
(DOCX)

**S4 Table. Results from additional analyses testing the hypotheses with inclusion of interaction effects of country and exposure variables (only reported for outcome 3—Hypothesis 2).**
(DOCX)

**S5 Table. Results from sensitivity analyses testing the effect of engaging in training (per-protocol analyses).**
(DOCX)

## Acknowledgments

We would like to recognise the valuable input of the members of the wider SCALA consortium to this work. We would also like to express our thanks to the patient and primary health care providers in Colombia, Mexico and Peru that took part in our study and made the research possible.

## Author Contributions

**Conceptualization:** Amy O'Donnell, Bernd Schulte, Jakob Manthey, Christiane Sybille Schmidt, Marina Piazza, Ines Bustamante Chavez, Guillermina Natera, Juliana Mejía-Trujillo, Augusto Pérez-Gómez, Antoni Gual, Hein de Vries, Adriana Solovei, Dasa Kokole, Eileen Kaner, Jurgen Rehm, Peter Anderson, Eva Jané-Llopis.

**Data curation:** Jakob Manthey, Jurgen Rehm, Peter Anderson, Eva Jané-Llopis.

**Formal analysis:** Bernd Schulte, Jakob Manthey, Christiane Sybille Schmidt, Jurgen Rehm.

**Funding acquisition:** Amy O'Donnell, Bernd Schulte, Jakob Manthey, Marina Piazza, Ines Bustamante Chavez, Guillermina Natera, Juliana Mejía-Trujillo, Augusto Pérez-Gómez, Antoni Gual, Hein de Vries, Eileen Kaner, Jurgen Rehm, Peter Anderson, Eva Jané-Llopis.

**Investigation:** Marina Piazza, Ines Bustamante Chavez, Guillermina Natera, Natalia Bautista Aguilar, Graciela Yazmín Sánchez Hernández, Juliana Mejía-Trujillo, Augusto Pérez-Gómez, Jurgen Rehm.

**Methodology:** Amy O'Donnell, Bernd Schulte, Jakob Manthey, Antoni Gual, Hein de Vries, Dasa Kokole, Eileen Kaner, Jurgen Rehm, Peter Anderson, Eva Jané-Llopis.

**Project administration:** Marina Piazza, Ines Bustamante Chavez, Guillermina Natera, Juliana Mejía-Trujillo, Augusto Pérez-Gómez, Peter Anderson, Eva Jané-Llopis.

**Supervision:** Bernd Schulte, Marina Piazza, Guillermina Natera, Augusto Pérez-Gómez, Jurgen Rehm, Peter Anderson, Eva Jané-Llopis.

**Writing – original draft:** Amy O'Donnell, Bernd Schulte, Jakob Manthey, Christiane Sybille Schmidt.

**Writing – review & editing:** Amy O'Donnell, Bernd Schulte, Jakob Manthey, Christiane Sybille Schmidt, Marina Piazza, Ines Bustamante Chavez, Guillermina Natera, Natalia Bautista Aguilar, Graciela Yazmín Sánchez Hernández, Juliana Mejía-Trujillo, Augusto Pérez-Gómez, Antoni Gual, Hein de Vries, Adriana Solovei, Dasa Kokole, Eileen Kaner, Carolin Kilian, Jurgen Rehm, Peter Anderson, Eva Jané-Llopis.

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
