## [Decision Letter · Decision Letter 0]

5 Jun 2021

PONE-D-21-06289

Primary care-based screening and management of depression amongst heavy drinking patients: Secondary outcomes of a three-country quasi-experimental study in Latin America

PLOS ONE

Dear Dr. O'Donnell, 

Thank you for submitting your manuscript to PLOS ONE. After careful consideration, we feel that it has merit but does not fully meet PLOS ONE’s publication criteria as it currently stands. Therefore, we invite you to submit a revised version of the manuscript that addresses the points raised during the review process.

The two reports have useful comments and I hope you will address them in the revised manuscript.

We look forward to receiving your revised manuscript.

Kind regards,

Santosh Kumar, Ph.D.

Academic Editor

PLOS ONE

**Santosh Kumar**

Associate Professor of Economics

Department of Economics and International Business

College of Business Administration

Sam Houston State University, Huntsville, TX, USA

**P**: 001 (936) 294 2416; **F**: 001 (936) 294 3488

**Email**: skumar@shsu.edu

Academic Editor: PLOS ONE

Research Fellow, Global Labor Organization (GLO)

Research Fellow, IZA Institute for Labor Economics (IZA, Bonn)

**Webpage**: https://sites.google.com/site/santoshkumar2987/

Journal Requirements:

2. Please include a separate caption for each figure in your manuscript.

Reviewers' comments:

Reviewer's Responses to Questions

**Comments to the Author**

1. Is the manuscript technically sound, and do the data support the conclusions?

Reviewer #1: Partly

Reviewer #2: Yes

2. Has the statistical analysis been performed appropriately and rigorously? 

Reviewer #1: N/A

Reviewer #2: Yes

3. Have the authors made all data underlying the findings in their manuscript fully available?

Reviewer #1: No

Reviewer #2: Yes

4. Is the manuscript presented in an intelligible fashion and written in standard English?

Reviewer #1: Yes

Reviewer #2: Yes

5. Review Comments to the Author

Reviewer #1: It is an interesting proposal for mental health issues in public health.

Title must be modify since authors presented partial, preliminary results because restrictions due to COVID 19. It was not possible to determine the community support on depression activity rates. In the same line, was not completed the evaluation on clinical intervention package (less versus more intensive) for screening and advice delivery by PHC.

Introduction must inform of similar studies and their findings in LIC and HIC so it si possible to compare and discuss afterwards.

The design story must tell that this survey is part of the international SCALA project to better understand the characteristics of municipal areas, even more if they were not randomly selected.

It would be important to detail providers profession and their training in depression since PHQ-2 don't evaluate suicide risk. In this point it would be interesting to clarify what happened with 83 persons that didn't receive treatment (from 615 heavy drinking persons, 230 qualify for depression too. From these 230, 147 (63,9%) received treatment, so were left 83 persons with out treatment?)

In the discuss section, would be interesting to contrast heavy drinking + depression findings 889.4%) with other results in the literature.

Reviewer #2: This is a very nicely written paper describing implementation of alcohol and depression screening in primary care in 3 Latin American countries. This topic is of high importance given the generally low rates of uptake of alcohol and depression screening in routine practice. The authors also offer compelling next steps and future directions. I would suggest adding a bit more detail in a few places as outlined in my specific comments below.

1) There is a large body of literature on collaborative care for depression as well as implementation of alcohol screening and intervention. It would be important to situate this study more in that literature in both the introduction and discussion sections.

2) The field of implementation science offers multiple implementation frameworks as well as discrete implementation strategies to improve uptake of health services. How does this study connect to that broader literature? Do your intervention strategies at the municipal and clinical levels draw from this literature?

3) The hypotheses are not presented until far into the method section. It would help the reader to have those presented earlier, such as at the end of the introduction section.

4) On page 9, it would be helpful for the authors to elaborate more on this sentence “due to the COVID-19 pandemic, only limited implementation of these measures was possible”. This would be a major factor in testing hypotheses 1 and 3. While there is mention of it in the limitations section, it would help the reader to better understand this early on in order to fully assess the results.

5) Booster sessions were mentioned in the intervention package section (pages 8-9). What did these entail?

6. PLOS authors have the option to publish the peer review history of their article (what does this mean?). If published, this will include your full peer review and any attached files.

Reviewer #1: No

Reviewer #2: No

---

## [Author Response · Author response to Decision Letter 0]

14 Jul 2021

Please see attached cover letter and table outlining our response to all reviewers' comments.

---

## [Editor Report · Decision Letter 1]

21 Jul 2021

PONE-D-21-06289 - Primary care-based screening and management of depression amongst heavy drinking patients: Interim secondary outcomes of a three-country quasi-experimental study in Latin America

PONE-D-21-06289R1

Dear Dr. Amy O'Donnell,

We’re pleased to inform you that your manuscript has been judged scientifically suitable for publication and will be formally accepted for publication once it meets all outstanding technical requirements.

Kind regards,

Santosh Kumar

Academic Editor

PLOS ONE
---

## [Editor Report · Acceptance letter]

26 Jul 2021

PONE-D-21-06289R1 

Primary care-based screening and management of depression amongst heavy drinking patients: Interim secondary outcomes of a three-country quasi-experimental study in Latin America 

Dear Dr. O'Donnell:

I'm pleased to inform you that your manuscript has been deemed suitable for publication in PLOS ONE. Congratulations! Your manuscript is now with our production department. 

Kind regards, 

on behalf of

Dr. Santosh Kumar 

Academic Editor

PLOS ONE